# ON PREDICTING GENERALIZATION USING GANS

Yi Zhang[1,2], Arushi Gupta[1], Nikunj Saunshi[1], and Sanjeev Arora[1]

[1]Princeton University, Computer Science Department
{y.zhang, arushig, nsaunshi, arora}@cs.princeton.edu
[2]Microsoft Research

## ABSTRACT

Research on generalization bounds for deep networks seeks to give ways to predict test error using just the training dataset and the network parameters. While generalization bounds can give many insights about architecture design, training algorithms etc., what they do not currently do is yield good predictions for actual test error. A recently introduced Predicting Generalization in Deep Learning competition (Jiang et al., 2020) aims to encourage discovery of methods to better predict test error. The current paper investigates a simple idea: can test error be predicted using *synthetic data,* produced using a Generative Adversarial Network (GAN) that was trained on the same training dataset? Upon investigating several GAN models and architectures, we find that this turns out to be the case. In fact, using GANs pre-trained on standard datasets, the test error can be predicted without requiring any additional hyper-parameter tuning. This result is surprising because GANs have well-known limitations (e.g. mode collapse) and are known to not learn the data distribution accurately. Yet the generated samples are good enough to substitute for test data. Several additional experiments are presented to explore reasons why GANs do well at this task. In addition to a new approach for predicting generalization, the counter-intuitive phenomena presented in our work may also call for a better understanding of GANs' strengths and limitations.

## 1 INTRODUCTION

Why do vastly overparametrized neural networks achieve impressive generalization performance across many domains, with very limited capacity control during training? Despite some promising initial research, the mechanism behind generalization remains poorly understood. A host of papers have tried to adapt classical generalization theory to prove upper bounds of the following form on the difference between training and test error:

$$\text{test} - \text{train} \leq \sqrt{\frac{C}{|S|}} + \text{tiny term}$$

where $S$ is the training dataset and $C$ is a so-called *complexity measure*, typically involving some function of the training dataset as well as the trained net parameters (e.g., a geometric norm). Current upper bounds of this type are loose, and even vacuous. There is evidence that such classically derived bounds may be too loose (Dziugaite & Roy, 2017) or that they may not correlate well with generalization (Jiang et al., 2019). This has motivated a more principled empirical study of the effectiveness of generalization bounds. The general idea is to use machine learning to determine which network characteristics promote good generalization in practice and which do not —in other words, treat various deep net characteristics/norms/margins etc. as inputs to a machine learning model that uses them to predict the generalization error achieved by the net. This could help direct theorists to new types of complexity measures and motivate new theories.

A recently started competition of Predicting Generalization in Deep Learning (Jiang et al., 2020) (PGDL) seeks to increase interest in such investigations, in an effort to uncover new and promising network characteristics and/or measures of network complexity that correlate with good generalization. As required in Jiang et al. (2020), a complexity measure should depend on the trained model,

optimizer, and training set, but not the held out test data. The first PGDL competition in 2020 did uncover quite a few measures that seemed to be predictive of good generalization but had not been identified by theory work.

**In this paper,** we explore a very simple baseline for predicting generalization that had hitherto not received attention: train a Generative Adversarial Network (GAN) on the training dataset, and use performance on the synthetic data produced by the GAN to predict generalization. At first sight GANs may not appear to be an obvious choice for this task, due to their well known limitations. For instance, while the goal of GANs training is to find a generator that fools the discriminator net —in the sense that the discriminator has low probability of spotting a difference between GAN samples and the samples from the true distribution—in practice the discriminator is often able to discriminate very well at the end, demonstrating that it was not fooled. Also, GANs' generators are known to exhibit mode collapse i.e., the generated distribution is a tiny subset of the true distribution. There is theory and experiments suggesting this may be difficult to avoid (Arora et al., 2018; Santurkar et al., 2018; Bau et al., 2019).

Given all these negative results about GANs, the surprising finding in the current paper is that GANs *do* allow for good estimates of test error (and generalization error). This is verified for families of well-known GANs and datasets including primarily CIFAR-10/100, Tiny ImageNet and well-known deep neural classifier architectures. In particular, in Section 3.1 and 3.2 we evaluate on the PGDL and DEMOGEN benchmarks of predicting generalization and present strong results. In Section 4 and 5, we also investigate reasons behind the surprising efficacy of GANs in predicting generalization as well as the effects of using data augmentation during GAN training.

## 2 RELATED WORK

**Generalization Bounds** Traditional approaches to predict generalization construct a generalization bound based on some notion of capacity such as parameter count, VC dimension, Rademacher complexity, etc. Neyshabur et al. (2018) provide a tighter generalization bound that decreases with increasing number of hidden units. Dziugaite & Roy (2017) reveal a way to compute nonvacuous PAC-Bayes generalization bounds. Recently, bounds based on knowledge distillation have also come to light (Hsu et al., 2020). Despite progress in these approaches, a study conducted by Jiang et al. (2019) with extensive hyper-parameter search showed that current generalization bounds may not be effective, and the root cause of generalization remains elusive. Given the arduous nature of constructing such bounds, an interest in complexity measures has arisen.

**Predicting Generalization in Deep Learning** The PGDL competition (Jiang et al., 2020) was held in NeuRIPS 2020 in an effort to encourage the discovery of empirical generalization measures following the seminal work of Jiang et al. (2018). The winner of the PGDL competition Natekar & Sharma (2020) investigated properties of representations in intermediate layers to predict generalization. Kashyap et al. (2021), the second place winner, experiment with robustness to flips, random erasing, random saturation, and other such natural augmentations. Afterwards, Jiang et al. (2021) interestingly find that generalization can be predicted by running SGD on the same architecture multiple times, and measuring the disagreement ratio between the different resulting networks on an unlabeled test set. There are also some pitfalls in predicting generalization, as highlighted by Dziugaite et al. (2020). They find that distributional robustness over a family of environments is more applicable to neural networks than straight averaging.

**Limitations of GANs** Arora et al. (2017) prove the lack of diversity enforcing in GAN's training objective, and Arora et al. (2018) introduce the Birthday Paradox test to conclude that many popular GAN models in practice do learn a distribution with relatively small support. Santurkar et al. (2018) and Bau et al. (2019) investigate the extent to which GAN generated samples can match the distributional statistics of the original training data, and find that they have significant shortcomings. Ravuri & Vinyals (2019) find that GAN data is of limited use in training ResNet models, and find that neither inception score nor FID are predictive of generalization performance. Notably, despite the small support, Arora et al. (2018) reveal that GANs generate distinct images from their nearest neighbours in the training set. Later, Webster et al. (2019) use latent recovery to conclude more carefully that GANs do not memorize the training set.

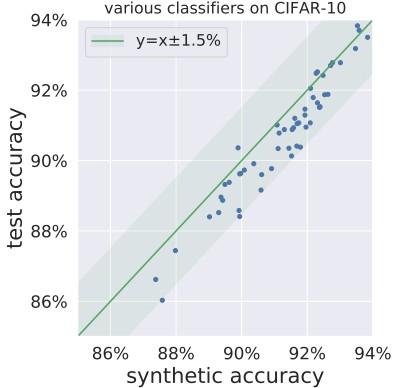 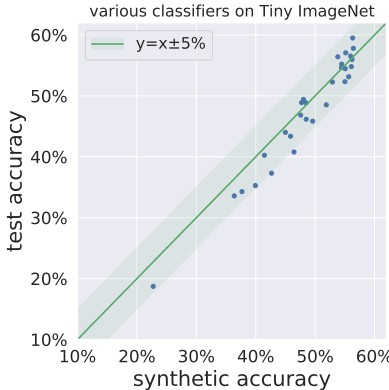

Figure 1: Scatter plots of test accuracy $g(f)$ v.s. synthetic accuracy $\hat{g}(f)$ with $f$ from a pool of deep net classifiers on CIFAR-10 and Tiny ImageNet of diverse architectures VGG, ResNet, DenseNet, ShuffleNet, NASNet, MobileNet with various hyper-parameters. One single BigGAN+DiffAug model is used for each dataset. **Left**: CIFAR-10 classifiers. The $y = x$ fit has $R^2$ score 0.804 and Kendall $\tau$ coefficient 0.851. **Right**: Tiny ImageNet classifiers. The $y = x$ fit has $R^2$ score 0.918 and Kendall $\tau$ coefficient 0.803.

**Theoretical Justification for Using GANs**  Arora et al. (2017) construct a generator that passes training against any bounded capacity discriminator but can only generate a small number of distinct data points either from the true data distribution or simply from the training set. For predicting generalization, it is crucial for the generator *not* to memorize training data. While Arora et al. (2017) do not answer why GANs do not memorize training data, a recent empirical study by Huang et al. (2021) demonstrates the difficulty of recovering input data by inverting gradients. Their work may cast light on how the generator could learn to generate data distinct from the training set when trained with gradient feedbacks from the discriminator. However, we are not aware of any theory that fully explains GANs' strength for predicting generalization despite limitations.

## 3 PREDICTING TEST PERFORMANCE USING GAN SAMPLES

We now precisely define what it means to predict test performance in our setting. We denote by $S_{\text{train}}, S_{\text{test}}$ and $S_{\text{syn}}$ the training set, test set and the synthetic dataset generated by GANs. Given a classifier $f$ trained on the training set $S_{\text{train}}$, we aim to predict its classification accuracy $g(f) := \frac{1}{|S_{\text{test}}|} \sum_{(x,y) \in S_{\text{test}}} \mathbf{1}\left[f(x) = y\right]$ on a test set $S_{\text{test}}$. Our proposal is to train a *conditional* GAN model on the very same training set $S_{\text{train}}$, and then sample from the generator a synthetic dataset $S_{\text{syn}}$ of labeled examples. In the end, we simply use $f$'s accuracy on the synthetic dataset as our prediction for its test accuracy. Algorithm 1 formally describes this procedure.

---

**Algorithm 1** Predicting test performance

---

**Require:** target classifier $f$, training set $S_{\text{train}}$, GAN training algorithm $\mathcal{A}$

  1. Train a conditional GAN model using $S_{\text{train}}$:

$$G, D = \mathcal{A}(S_{\text{train}}) \text{ where } G, D \text{ are the generator and discriminator networks.}$$

  2. Generate a synthetic dataset by sampling from the generator $G$:

$$S_{\text{syn}} = \{(\tilde{x}_1, \tilde{y}_1), \ldots, (\tilde{x}_N, \tilde{y}_N)\} \text{ where } \tilde{x}_i, \tilde{y}_i = G(z_i, \tilde{y}_i).$$

  The $z_i$'s are drawn i.i.d. from $G$'s default input distribution. $N$ and $\tilde{y}_i$' are chosen so as to match statistics of the training set.

**Output:** the synthetic accuracy $\hat{g}(f) := \frac{1}{|S_{\text{syn}}|} \sum_{(\tilde{x}, \tilde{y}) \in S_{\text{syn}}} \mathbf{1}\left[f(\tilde{x}) = \tilde{y}\right]$ as the prediction

---

**Remark.** *Any $N \geq |S_{train}|$ is a safe choice to ensure that $\hat{g}(f)$ concentrates[1] around its mean $\mathbb{E}_{z,\tilde{y}}\left[\mathbf{1}\left[f(G(z, \tilde{y})) = \tilde{y}\right]\right]$ and its deviation has negligible influence on the performance.*

---

[1]the deviation is only $O\left(1/\sqrt{N}\right)$ by standard concentration bounds

| Task | No.1 team | | | No.2 team | No.3 team | Ours |
|------|-----------|---|---|-----------|-----------|------|
| | DBI*LWM | MM | AM | R2A | VPM | |
| 1 : VGG on CIFAR-10 | 25.22 | 1.11 | 15.66 | 52.59 | 6.07 | **62.62** |
| 2 : NIN on SVHN | 22.19 | 47.33 | **48.34** | 20.62 | 6.44 | 34.72 |
| 4 : AllConv on CINIC-10 | 31.79 | 43.22 | 47.22 | **57.81** | 15.42 | 52.80 |
| 5 : AllConv on CINIC-10 | 15.92 | 34.57 | 22.82 | 24.89 | 10.66 | **53.56** |
| 8 : VGG on F-MNIST | 9.24 | 1.48 | 1.28 | 13.79 | 16.23 | **30.25** |
| 9 : NIN on CIFAR-10 | 25.86 | 20.78 | 15.25 | 11.30 | 2.28 | **33.51** |

Table 1: Comparison of our method to the Top-3 teams of the PGDL competition. Scores shown are calculated using the Conditional Mutual Information metric, higher is better. DBI*LWM=Davies-Bouldin Index*Label-wise Margin, MM=Mixup Margin and AM=Augment Margin are the proposed methods from first place solution by Natekar & Sharma (2020). R2A=Robustness to Augmentations (Aithal et al., 2021) and VPM=Variation of the Penultimate layer with Mixup (Lassance et al., 2020) are the second and third place solutions. Task 4 and Task 5 differ in that Task 4 classifiers have batch-norm while Task 5 do not.

We demonstrate the results in Figure 1. The test accuracy $g(f)$ consistently resides in a small neighborhood of $\hat{g}(f)$ for a diverse class of deep neural net classifiers trained on different datasets. For the choice of GAN architecture, we adopt the pre-trained BigGAN+DiffAug models (Brock et al., 2019; Zhao et al., 2020) from the StudioGAN library[2]. We evaluate on pre-trained deep neural net classifiers with architectures ranging from VGG-(11, 13, 19), ResNet-(18, 34, 50), DenseNet-(121, 169), ShuffleNet, PNASNet and MobileNet trained on CIFAR-10 and Tiny ImageNet with hyper-parameter setting (`learning rate`, `weight decay factor`, `learning rate schedule`) uniformly sampled from the grid $\left\{10^{-1}, 10^{-2}\right\} \times \left\{5 \times 10^{-4}, 10^{-3}\right\} \times \{$`CosineAnnealing`, `ExponentialDecay`$\}$. We use SGD with momentum 0.9 and batch size 128 and data augmention of horizontal flips for training all classifiers.

### 3.1 EVALUATION ON PGDL COMPETITION

We evaluate our proposed method on the tasks of NeurIPS 2020 Competition on Predicting Generalization of Deep Learning. The PGDL competition consists of 8 tasks, each containing a set of pre-trained deep net classifiers of similar architectures but with different hyper-parameter settings, as well as the training data. The tasks of PGDL are based on a wide range of datasets inlcuding CIFAR-10, SVHN, CINIC-10, Oxford Flowers, Oxford Pets and Fashion MNIST. For every task, the goal is to compute a scalar prediction for each classifier based on its parameters and the training data that *correlates* as much as possible with the actual generalization errors of the classifiers measured on a test set. The correlation score is measured by the so-called Conditional Mutual Information, which is designed to indicate whether the computed predictions contain all the information about the generalization errors such that knowing the hyper-parameters does not provide additional information, see (Jiang et al., 2020) for details.

Since the goal of PGDL is to predict generalization gap instead of test accuracy, our proposed solution is a slight adaptation of Algorithm 1. For each task, we first train a GAN model on the provided training dataset and sample from it a labeled synthetic dataset. Then for each classifier within the task, instead of the raw synthetic accuracy, we use as prediction the gap between training and synthetic accuracies. For all tasks, we use BigGAN+diffAug with the default[3] implementation for CIFAR-10 from the StudioGAN library. We found that a subset of them (Task 6 with Oxford Flowers and Task 7 with Oxford Pets) were not well-suited for standard GAN training without much tweaking due to a small training set and highly uneven class sizes. We focused only on the subset of the tasks where GAN training worked out of the box, specifically Task 1, 2, 4, 5, 8, and 9.

We report the results in Table 1, where we observe that on tasks involving either CIFAR-10 or VGG-like classifiers, our proposed solution out-performs all of the solutions from the top 3 teams of the competition by a large margin. One potential reason may be that the default hyper-parameters of

---

[2]available at https://github.com/POSTECH-CVLab/PyTorch-StudioGAN

[3]Task 8 uses the single-channel $28 \times 28$ Fashion-MNIST dataset. For GAN training, we symmetrically zero-pad the training images to $32 \times 32$ and convert to RGB by replicating the channel. To compute the synthetic accuracy, we convert the generated images back to single-channel and perform a center cropping.

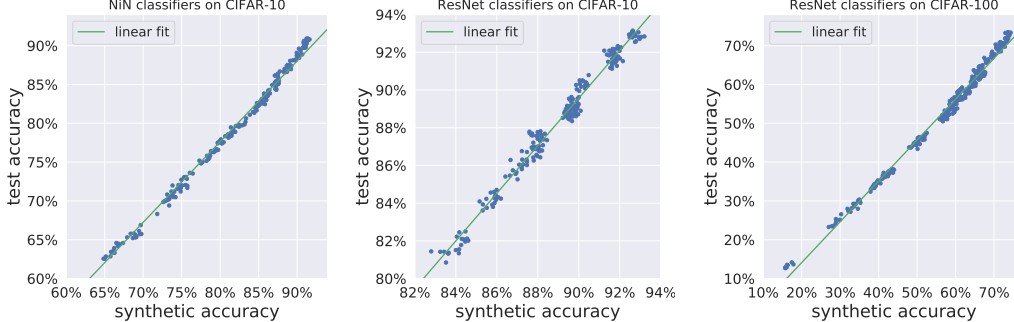

Figure 2: Scatter plots of test accuracy $g(f)$ v.s. synthetic accuracy $\hat{g}(f)$ with $f$ from DEMOGEN consisting of 216 Network in Network classifiers and 216 ResNet classifiers on CIFAR-10 as well as 324 ResNet classifiers on CIFAR-100. The synthetic examples are from the BigGAN+DiffAug.

the BigGAN model have been engineered towards CIFAR-10 like datasets and VGG/ResNet-like discriminator networks. It is worth mentioning that we conducted *absolutely zero* hyper-parameter search for all tasks here. Especially for CIFAR-10 tasks, we directly take the pre-trained models from the StudioGAN library. It is likely that we may achieve even better scores with fine-tuned GAN hyper-parameters and optimized training procedure for each PGDL task, and we leave it to future work.

## 3.2 EVALUATION ON DEMOGEN BENCHMARK

The seminal work of Jiang et al. (2018) constructed the Deep Model Generalization benchmark (DEMOGEN) that consists of 216 ResNet-32 (He et al., 2016) models and 216 Network-in-Networks (Lin et al., 2013) models trained on CIFAR-10 plus another 324 ResNet-32 models trained on CIFAR-100, along with their training and test performances. These models are configured with different normalization techniques (group-norm v.s. batch-norm), network widths, learning rates, weight-decay factors, and notably whether to train with or without data augmentation.

The goal of DEMOGEN, different from PGDL, is to *numerically* predict generalization gap (or equivalently test accuracy) with the additional knowledge of the test accuracies of a few pre-trained classifiers. Following the setup of Jiang et al. (2018), we use a simple linear predictor $g(f) = a \cdot \hat{g}(f) + b$ with two scalar variables $a, b \in \mathbb{R}$ instead of directly using $\hat{g}(f)$ for the best performance. Here $a, b$ are estimated using least squares regression on a separate pool of pre-trained classifiers with known test accuracies. Jiang et al. (2018) proposes to use variants of coefficient of determination ($R^2$) of the resulted predictor as evaluation metrics. Specifically, the adjusted $R^2$ and $k$-fold $R^2$ are adopted (see Jiang et al. (2018) for concrete definitions).

| | NiN+CIFAR-10 | | ResNet+CIFAR-10 | | ResNet+CIFAR-100 | |
|---|---|---|---|---|---|---|
| | adj $R^2$ | 10-fold $R^2$ | adj $R^2$ | 10-fold $R^2$ | adj $R^2$ | 10-fold $R^2$ |
| `qrt+log` | 0.94 | 0.90 | 0.87 | 0.81 | 0.97 | 0.96 |
| `qrt+log+unnorm` | 0.89 | 0.86 | 0.82 | 0.74 | 0.95 | 0.94 |
| `moment+log` | 0.93 | 0.87 | 0.83 | 0.74 | 0.94 | 0.92 |
| Self-Attention GAN | 0.992 | 0.991 | 0.903 | 0.897 | — | — |
| Spec-Norm GAN | 0.994 | 0.993 | 0.874 | 0.869 | — | — |
| BigGAN | 0.988 | 0.986 | 0.911 | 0.904 | 0.992 | 0.991 |
| BigGAN+ADA | 0.991 | 0.989 | 0.932 | 0.928 | — | — |
| BigGAN+DiffAug | 0.995 | 0.995 | 0.981 | 0.921 | 0.993 | 0.993 |

Table 2: Comparison of predicting generalization using GAN samples to the Top-3 methods of Jiang et al. (2018) (`qrt+log`, `qrt+log+unnorm` and `moment+log`) on DEMOGEN. Overall, our method of predicting generalization using GAN samples outperforms any of the previous solution by a large margin. The BigGAN+DiffAug appears to be the best performing. '—' means not tested.

In Table 2, we compare the scores of our method with several state-of-the-art GAN models. Specifically, we again take the pre-trained GAN models from the StudioGAN library to generate synthetic datasets for the CIFAR-10 tasks. For the CIFAR-100 task, we use the same hyper-parameters as the CIFAR-10 models to train a plain BigGAN and a BigGAN+DiffAug model from scratch. Again, each synthetic dataset is sampled only once. Overall, predicting using GAN samples achieves remarkably higher scores than all methods proposed in Jiang et al. (2018). We also visualize the resulted linear fit for each task in Figure 2 where we observe the strong linear correlation. Note that the resulting linear fits are close the ideally desired $y = x$ fit. In fact, using directly $\hat{g}(f)$ as the prediction would also lead to extremely competitive $R^2$ values 0.928, 0.995 and 0.975 respectively.

## 4 GAN SAMPLES ARE CLOSER TO TEST SET THAN TRAINING SET

In this section, we demonstrate a surprising property of the synthetic examples produced by GANs that may provide insights into their observed ability to predict generalization: synthetic datasets generated by GANs are 'closer' to test set than training set in the feature space of well-trained deep net classifiers, even though GAN models had (indirect) access to the training set but no access to the test set. Here 'well-trained' classifiers are the ones that almost perfectly classify the training examples, i.e. its training accuracy is $> 97\%$. To define closeness properly, we modify the widely used standard Fréchet Distance proposed by Heusel et al. (2017) to account for labeled examples. We name our modified version as the Class-conditional Fréchet Distance.

### 4.1 CLASS-CONDITIONAL FRÉCHET DISTANCE

The goal is to measure how close the *labeled* synthetic dataset is to the test set in the eyes of a certain pre-trained classifier. However, FID does not capture any label information by default. Our proposed class-conditional Fréchet Distance can be viewed as the sum over each class of the Fréchet Distance between test and synthetic examples in a certain feature space. The original FID measures a certain distance between the feature embeddings of real examples and synthetic examples from GANs, in the feature space of an Inception network pre-trained on ImageNet. For our purposes, we modify the notion of FID to use any feature extractor $h$. Specifically, we define the distance between two sets of examples $S$ and $S'$ w.r.t. a feature extractor $h$ as

$$\mathcal{F}_h\left(S, S'\right) := \|\mu_S(h) - \mu_{S'}(h)\|_2^2 + \mathrm{Tr}\left[\mathrm{Cov}_S(h) + \mathrm{Cov}_{S'}(h) - 2\left[\mathrm{Cov}_S(h)\,\mathrm{Cov}_{S'}(h)\right]^{1/2}\right]$$

where we have the empirical mean of $h$'s output $\mu_S(h) := \frac{1}{|S|}\sum_{(x,y)\in S} h(x)$ and the empirical covariance $\mathrm{Cov}_S(h) := \frac{1}{|S|}\sum_{(x,y)\in S}(h(x) - \mu_S(h))(h(x) - \mu_S(h))^\top$ over the examples from $S$, and $\mu_{S'}(h), \mathrm{Cov}_{S'}(h)$ are computed analogously. Furthermore, $[\cdot]^{1/2}$ is the matrix square root and $\mathrm{Tr}[\cdot]$ is the trace function.

For labeled datasets $S$ and $\tilde{S}$, we denote by $\mathcal{C}$ the collection of all possible labels. For each label $c \in \mathcal{C}$, we let $S_c$ and $\tilde{S}_c$ denote the the subsets of examples with label $c$ in $S$ and $\tilde{S}$ respectively. This leads to the class-conditional Fréchet Distance:

$$d_h\left(S, \tilde{S}\right) := \sum_{c\in\mathcal{C}} \mathcal{F}_h\left(S_c, \tilde{S}_c\right)$$

In Figure 3, we evaluate $d_h$ on CIFAR-10 among the training set $S_{\mathrm{train}}$, the test set $S_{\mathrm{test}}$ and the synthetic set $S_{\mathrm{syn}}$, with $h$ being the feature map immediately before the fully-connected layer of each pre-trained deep net used in the middle sub-figure of Figure 2. The synthetic dataset is generated by the same BigGAN+DiffAug model used in Figure 2 as well. We observe that for all pre-trained nets, $d_h(S_{\mathrm{syn}}, S_{\mathrm{test}}) < d_h(S_{\mathrm{train}}, S_{\mathrm{test}})$ by a big margin, which provides a partial explanation for why GAN synthesized samples may be a good replace for the test set. Furthermore, for every well-trained net, $d_h(S_{\mathrm{syn}}, S_{\mathrm{test}}) < d_h(S_{\mathrm{syn}}, S_{\mathrm{train}})$ as well. This is particularly surprising because $S_{\mathrm{syn}}$ ends up being 'closer' to the *unseen* $S_{\mathrm{test}}$ rather than $S_{\mathrm{train}}$, despite the GAN model being trained to minimize certain distance between $S_{\mathrm{syn}}$ and $S_{\mathrm{train}}$ as part of the training objective.

### 4.2 CLASS-CONDITIONAL FRÉCHET DISTANCE ACROSS CLASSIFIER ARCHITECTURES

It is natural to wonder whether we can attribute the phenomenon of GAN samples being closer to test set to the architectural similarity between the evaluated classifiers and the discriminator network

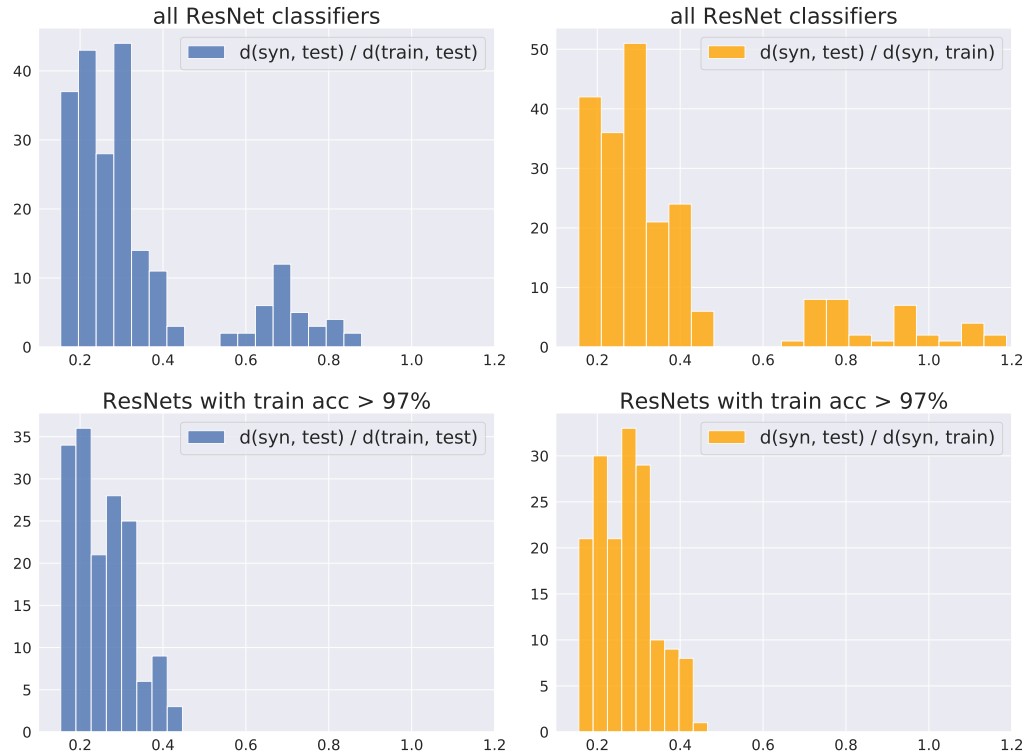

Figure 3: Histograms of $d_h$'s measured using the feature spaces of the pool of 216 pre-trained ResNet models on CIFAR-10 in DEMOGEN. Since features vectors $h(x)$ of different classifiers may have vastly different norms, we plot the histograms of the ratios $d_h(S_{\text{syn}}, S_{\text{test}})/d_h(S_{\text{train}}, S_{\text{test}})$ and $d_h(S_{\text{syn}}, S_{\text{test}})/d_h(S_{\text{syn}}, S_{\text{train}})$ instead. The synthetic dataset is the identical one generated by the BigGAN-DiffAug model used in Section 3.2. **Top**: the histograms of all the 216 pre-trained classifiers, including the not-so-well-trained ones with training accuracy below 90%. **Bottom**: the histograms of only the well-trained 162 out of 216 classifiers with training accuracy at least 97%. Clearly, the ratio $d_h(S_{\text{syn}}, S_{\text{test}})/d_h(S_{\text{train}}, S_{\text{test}})$ is below 1 consistently, and the ratio $d_h(S_{\text{syn}}, S_{\text{test}})/d_h(S_{\text{train}}, S_{\text{test}})$ is below 1 for all well-trained classifiers. This indicates that in their feature space the labeled synthetic examples behave more similarly to the test set rather than to the training set.

used in the GAN model — they are all derivatives from the ResNet family. Thus we investigate the values of $d_h$ for the collection of classifiers with diverse architectures used in Figure 1, which contains network architectures that may be remote from ResNet such as VGG, MobileNet, PNASNet etc. The result is demonstrated in Figure 4, where we observe the surprising phenomenon that the single copy of the synthetic dataset is regarded closer to the test set by all classifiers in the collection. This suggests that certain GAN models are capable of matching, to a degree, the feature distribution universally for a diverse set of deep neural classifiers.

## 5 EXPLORATION: EFFECTS OF DATA AUGMENTATION

In this section, we discuss certain intriguing effects of data augmentation techniques used during GAN training on the performance of the resulted synthetic datasets for predicting generalization. We use the CIFAR-100 task from DEMOGEN as a demonstration where the effects are the most pronounced. The task consists of 324 ResNet classifiers, only half of which are trained with data augmentation, specifically random crop and horizontal flipping.

Given the success of complexity measures that verify 'robustness to augmentation' in the PGDL competition, one would imagine applying data augmentations to the real examples during GAN training could be beneficial for predicting generalization, since the generator would be encouraged

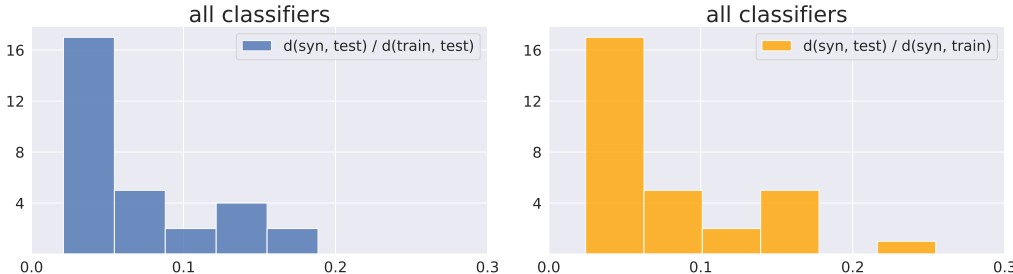

Figure 4: Histograms of $d_h$'s measured using the feature spaces of deep neural nets with various architectures on CIFAR-10 (same ones from Figure 1). The synthetic dataset is the same one used in Figure 1 as well. All the classifiers are well-trained. $d_h(S_{\text{syn}}, S_{\text{test}})$ is consistently the smallest among all three pairwise distances among $S_{\text{train}}, S_{\text{test}}$ and $S_{\text{syn}}$.

to match the distribution of the augmented images. However, we show in Figure 5 this is not the case here.

Specifically, Figure 5a indicates that, when evaluated on a synthetic dataset from a GAN model trained with augmentation only applied to real examples, the ResNets trained without any data augmentation exhibit a test accuracy v.s. synthetic accuracy curve that substantially deviates from $y = x$, while the ones trained with data augmentations are relatively unaffected. In comparison, once data augmentation is removed from GAN's training procedure, this deviation disappears, as captured in Figure 5b. This could be because the ResNets trained without data augmentation are sensitive to the perturbations learned by the generator. Overall, the best result is achieved by deploying the differential augmentation technique that applies the same differentiable augmentation to both real and fake images and enables the gradients to be propagated through the augmentation back to the generator so as to regularize the discriminator without manipulating the target distribution.

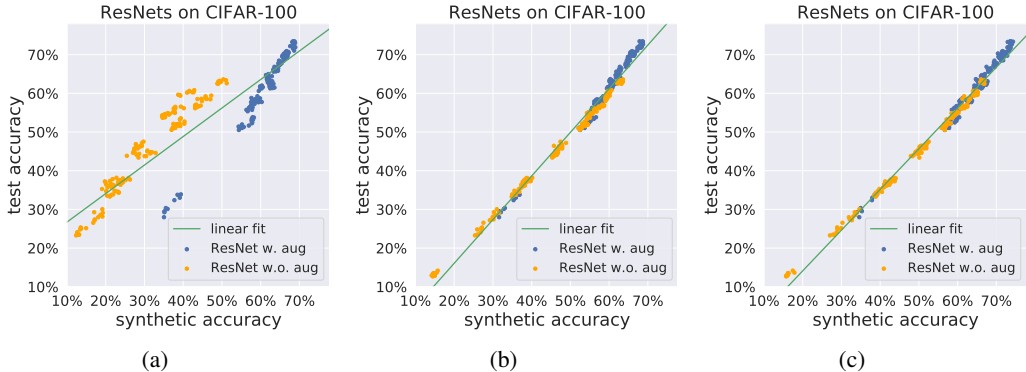

Figure 5: Scatter plots of test accuracy $g(f)$ v.s. synthetic accuracy $\hat{g}(f)$ on the DEMOGEN CIFAR-100 task using BigGAN models trained with **a)** data augmentation (random crop + horizontal Flipping) only to real (training) examples, **b)** without any data augmentation, and **c)** with the differentiable data augmentation technique. Blue and orange dots represent ResNet-34 classifiers trained with and without data augmentation.

## 6 DISCUSSION AND CONCLUSIONS

Competitions such as PGDL are an exciting way to inject new ideas into generalization theory. The interaction of GANs and good generalization needs further exploration. It is conceivable that the quality of the GAN-based prediction is correlated with the niceness of architectures in some way. It would not be surprising that it works for popular architectures but does not work for all. We leave it as future work to determine the limits of using GANs for predicting generalization.

Our finding presented here offers a new approach for predicting generalization, and it sheds light on a possibly bigger question —*what can GANs learn despite known limitations?* To our best knowledge, we are not aware of any existing theory that completely explains this counter-intuitive empirical phenomena. Such a theoretical understanding will necessarily require innovative techniques and thus is perceivably non-trivial. However, we believe it is an important future direction as it will potentially suggest more use cases of GANs and will help improve existing models in a principled way. We hope our work inspires new endeavors along this line.

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
