# OpenReview forum: "On Predicting Generalization using GANs"
_ICLR.cc/2022/Conference — ICLR 2022 Spotlight_

### Official Review · Reviewer_d8Mz · 2021-10-30

**Correctness:** 4
**Technical Novelty And Significance:** 3
**Empirical Novelty And Significance:** 3
**Recommendation:** 8
**Confidence:** 4

**Main Review:**

This paper proposes a simple method for estimating the generalization gap via training a GAN. The central idea of the paper is straightforward and the presentation of the idea is very clear. As far as the method is concerned, I do not have any concerns. The experimental results are quite compelling. In fact, I believe that the performance of the method might be good enough to be used in some production scenarios where training data is expensive and the main interest is to do model selection or estimate the population risk. The additional analysis on representations learned by GANs is also interesting since it offers another perspective on the common belief that GANs are bad at generating from the full data distribution (although FID does have many problems). A broader implication of the paper is that it suggests there is a subtle difference between generative modeling of the data and discriminative learning such as classification for high dimensional data.

The weaknesses are the following:
- the paper feels hastily put together. While the central idea and results are clear, the entire paper reads very unpolished. For example, some paragraphs do not have line breaks between them (e.g., the first paragraph of section 4) and it is unclear why there is a line break in the abstract. In related works, a lot of the bold points only have one or two sentences following them. There are also some extremely short paragraphs throughout the paper such as 2.1. I believe that many of these can be merged. In equation one, the introduction of uniform convergence could use a bit more formality. Citation styles are also not consistent. For example,  “(Dziugaite & Roy)” does not have a year in the citation. The two tables also do not have consistent styles.
- Maybe I missed this but I cannot find the number of synthetic data generated. Is it the same as the size of the true dataset? Along the same line, how does the estimated change as the function of different samples from the GAN? Specifically, I would be interested to see the variance coming from using different synthetic data from the GAN.
- What is special about GANs? There are many families of deep generative models. Is there a specific reason why GAN was chosen over the other models? If so, it would be nice to see some comparisons. If the other generative models do equally well, it would also be interesting.

If the authors can address my concerns, I will increase my score.

---------------------------

**Update**

I am happy with the authors' response and the changes made to the paper. As promised, I am increasing my score to 8.

**Summary Of The Paper:**

This paper investigates the task of using GAN’s samples as a substitute for the true test distribution for estimating the generalization. Experiments show that samples from GAN are a powerful substitute for the test set and the predictive power of the synthetic accuracy outperforms all prior methods include the winner of the 2020 PGDL competition, despite all the empirical and theoretical evidence that GANs are not good at approximating the true data distribution. The paper then performs a preliminary investigation on why samples from GANs are a good substitute for the test set and found that using a certain notion of perceptual similarity, samples from GAN’s are much closer to the test distribution than the training distribution despite being trained on the training data.

**Summary Of The Review:**

The paper offers a simple method for estimating the performance of deep models without test data using GANs. The empirical results are strong and the analysis is interesting. On the other hand, the paper is unpolished and the presentation could be improved.

---

> ### Author Response · Authors · 2021-11-22
> **Response to Reviewer d8Mz**
>
> We thank the reviewer for the positive evaluations and the valuable suggestions!
> We have revised the paper to fix the issues and to incorporate the suggestions, and we hope our response addresses your concerns.
>
> ### Q: “the entire paper reads very unpolished.”
> A: Thanks for the suggestions for improving our paper. They have been incorporated in the revision.
>
>
> ### Q: “how does the estimated change as the function of different samples from the GAN?”. “interested to see the variance coming from using different synthetic data from the GAN”
>
> A:
> Note that the number of synthetic samples is *not critical* to the performance as long as it is reasonably large (e.g. same size as the training set), since the synthetic accuracy concentrates tightly around its mean (within $\mathcal{O}(1/\sqrt{N})$) over the randomness of sampling the generator, by standard concentration bounds. As long as the size of the synthetic dataset is larger than the training/test set, the deviation of the synthetic accuracy will have little influence on the final score.
>
> We have revised the paper to emphasize that the synthetic datasets have the same size as their respective training sets throughout all experiments. We have also added a remark on why the number of synthetic samples is not critical.
>
>
>
>
> ### Q: Is there a specific reason why GAN was chosen over other generative models?
>
> A:
> In the revision, we have added some theoretical justification for using GANs.
>
> The most significant motivation to use GANs is that the samples from the generator have been overseen by a neural network classifier (the discriminator) during training, which is a distinct feature of GANs compared to other generative models such as VAE or flow-based models. Furthermore, most state-of-the-art generative models in terms of FID scores on CIFAR-10 are GAN models with very few exceptions. Lastly, in terms of the availability of online implementations, open-sourced GAN models most easily satisfy our requirement for being capable of generating *labeled* data, while VAE and flow-based models are not so frequently implemented in the conditional versions. For example, we attempted to look into the best performing non-GAN model on CIFAR-10, SBGM (https://arxiv.org/pdf/2106.05931v2.pdf) but then realized its online repository contains only unconditional models.
>
> That said, we believe it is an interesting future direction to explore the possibilities of other generative models for predicting generalization.

---

### Official Review · Reviewer_6YD8 · 2021-11-02

**Correctness:** 3
**Technical Novelty And Significance:** 3
**Empirical Novelty And Significance:** 3
**Recommendation:** 5
**Confidence:** 4

**Main Review:**

### Strengths:
1. The proposed outperforms existing methods on both PGDL and DEMOGEN benchmarks.
2. As far as I know, the proposed method is novel. Also, it is quite interesting and counterintuitive if it works.

### Weaknesses:
1. The most critical weakness of this paper is that it doesn't specify how many synthesized samples are used to compute the synthetic accuracy. It is not clear to the readers how to compute the synthetic accuracy and how many samples are enough. An analysis of the number of samples versus the performance metrics (e.g. Conditional Mutual Information metric) is also missing.
2. The authors do not provide any statistical analysis of the proposed method. For example, each experiment is only conducted once instead of providing the mean and standard deviation of multiple runs. It is not clear how the performance would be affected by the randomness in sampling the synthesized data or training GANs.
3. The reproducibility of the paper is questionable. The authors do not provide any hyperparameters and training details.
4. The proposed method only works for the image classification task. Moreover, the experiments are only conducted on balanced datasets. It is not clear whether it generalizes to unbalanced datasets such as iNaturalist.
5. The GAN models the authors use in the paper are descendants of SN-GAN which all use hinge loss, similar model architecture, conditional BatchNorm, and projection discriminator. It would be better to see a more diverse collection of GAN models being studied.


### Typos:
1. The year of the reference `Dziugaite & Roy` is missing.
2. On the first page, the citation format is inconsistent with the rest of the paper: `(Jiang et al. (2019))` should be `(Jiang et al., 2019)`.

**Summary Of The Paper:**

This paper proposes to use synthesized data generated by GANs to estimate the generalization of image classification models. The authors argue that the GAN samples are closer to the test set than the training set.

**Summary Of The Review:**

Overall, I think the proposed method is quite interesting but lacks well-conducted experiments and evidence to support it. Based on the current state of the paper, I give a weak reject; however, I am willing to increase the score if the authors can address the weaknesses and update their draft during the rebuttal period.

---

> ### Author Response · Authors · 2021-11-22
> **Response to Reviewer 6YD8**
>
> We thank the reviewer for the questions and for pointing out typos!
> We have revised the paper to fix the typos and we hope our response addresses your concerns.
>
> ### Q: “The most critical weakness of this paper is that it doesn't specify how many synthesized samples are used to compute the synthetic accuracy.”
>
> A: Note that the number of synthetic samples is *not critical* to the performance as long as it is reasonably large (e.g. same size as the training set), since the synthetic accuracy concentrates tightly around its mean (within $\mathcal{O}(1/\sqrt{N})$) over the randomness of sampling the generator, by standard concentration bounds.
>
> Throughout the paper, the synthetic datasets have the same size as their respective training sets. We have revised the paper to include this information as well as to explain why the number of synthetic samples is not critical.
>
>
> ### Q: “do not provide any statistical analysis of the proposed method, each experiment is only conducted once instead of providing the mean and standard deviation of multiple runs”. no mean and std provided. “not clear how the performance would be affected by the randomness in sampling the synthesized data or training GANs“
>
> A: As discussed above, the randomness due to sampling the synthetic dataset is not a big issue. As long as the size of the synthetic dataset is larger than the training/test set, the deviation of the synthetic accuracy will not influence the final score.
>
> As for the randomness of training GANs, we performed some extra experiments to show the std from multiple runs of GAN training is small. For example, the mean and std of the score of our method on PGDL Task 1 are 61.58 +- 1.25, over 5 independent runs of GAN training.
>
> Note that our statement of “each experiment is only conducted once” should be taken positively since it shows we are not exploiting randomness. This statement should not be confused with “conduct each experiment multiple times but only report the best”.
>
> The fact our method wins by a large margin despite being conducted once shows its reliability and reproducibility to a further extent.
>
>
> ### Q: “reproducibility of the paper is questionable. The authors do not provide any hyperparameters and training details“:
>
> A: We respectfully disagree with this. Just the opposite, we believe reproducibility is a strength of our method. As already emphasized in the paper, the GAN models used in all of our experiments are either from downloaded pre-trained weights or trained with default hyperparameters from the StudioGAN library available online (link included in the paper as well). We have performed *absolutely zero* hyper-parameter searches for any of the experiments.
>
> Also, as discussed above, “conducting each experiment once” further shows the reproducibility of our method.
>
>
> ### Q: “proposed method only works for the image classification task”. “the experiments are only conducted on balanced datasets, not clear whether it generalizes to unbalanced datasets such as iNaturalist”
>
> A: So far predicting generalization benchmarks (i.e. PGDL and DEMOGEN) and related previous works have been *exclusively* on image classification tasks. We are confident our method will perform well on any domain that GANs work well and we are more than happy to run experiments on any future predicting generalization benchmarks beyond image classification.
>
> Note that the reviewer’s assessment “the experiments are only conducted on balanced datasets” is inaccurate. The PGDL Task 2 is on SVHN, which is an *unbalanced* dataset.
>
> Admittedly, our method does not beat the previous winning solution on PGDL Task 2, and the reason may be possibly attributed to the unbalanced-ness. unbalanced-ness does pose particular challenges to GAN training, but there are many known techniques to tackle it, such as entropy weighted labels (https://ojs.aaai.org//index.php/AAAI/article/view/5142).
>
> As we emphasized in the paper, in the current work, we focus on demonstrating the power of off-the-shelf GAN models for predicting generalization, and “we leave it as a future work to determine the limits of using GANs for predicting generalization.”
>
>
>
>
> ### Q: “it would be better to see a more diverse collection of GAN models being studied.“
>
> A:
> This is definitely an interesting direction to move forward. We have some preliminary results that StyleGAN, which uses logistic loss instead of hinge loss, is also competitive on PGDL (i.e. scoring 58 on PGDL Task 1). Note that the goal of the current submission was not a comprehensive study of the strength and limitations of different GAN models but just a demonstration that popular GANs (which presumably are popular precisely because they appear to do better on distribution learning) can also be used to quantify test error.

---

> > ### Comment · Reviewer_6YD8 · 2021-11-29
> > **Thanks for your clarification. Increasing my rating to accept.**
> >
> > After reading the authors' response and the revised paper, I believe most of my concerns regarding the sample size and reproducibility have been addressed. Therefore, I would like to increase my rating to accept.

---

### Official Review · Reviewer_jdtm · 2021-11-02

**Correctness:** 4
**Technical Novelty And Significance:** 3
**Empirical Novelty And Significance:** 4
**Recommendation:** 8
**Confidence:** 4

**Main Review:**

##########################################################################

Pros:

* Predicting generalization is an timely and relevant problem.

* Empirical findings are strong.

* Appropriate empirical methodology is used (Jiang et al. 2020)

##########################################################################

Cons:

* The authors claim to "generalize the notion of FID" by applying it at different layers. It seems unnecessary to claim a generalization here, as its just a small modification to FID.

* No theoretical justifications are made for the observations. This is often the case for new empirical findings, but the paper would be stronger with some theory. I am left wondering if there are possible confounding factors which are leading to the observations.


##########################################################################

Questions during rebuttal period:

Any insight into why exactly GANs produce data more similar to the test set?


##########################################################################


Post-rebuttal:

I found the authors response to my question about theoretical justification interesting and persuasive and decided to raise my score.

**Summary Of The Paper:**

This work studies generalization prediction with synthetic data produced by GANs. Specifically the training accuracy gap between synthetic and given data is proposed as a measure for generalization. The authors show strong improvement over other PGDL and DEMOGEN results. A further empirical finding is made on the greater similarity between test and synthetic samples than test and train samples, as measured by FID. A variety of GAN architectures are explored.


**Summary Of The Review:**

Interesting and relevant empirical results on predicting generalization with a simple technique based on synthetic data generated by GANs.

---

> ### Author Response · Authors · 2021-11-22
> **Response to Reviewer jdtm**
>
> We thank the reviewer for the positive evaluations and the useful suggestions!
>
> ### C: Change “generalization of FID” to “modification to FID”.
> A: We appreciate the suggestion. It has been changed in the revision.
>
> ### Q: No theoretical justification is provided. Could there be confounding factors leading to this observation? Any insight into why exactly GANs produce data more similar to the test set than to the training set?
>
> A: We provide some theoretical justification on page 2 in the revision. In fact, the paper [Arora et al 2017] proves the possibility of a generator that has mode collapse and yet can be used to predict the generalization of any bounded capacity classifier, provided the generator does not memorize training data. In fact, the recent empirical work on the difficulty of gradient inversion (https://proceedings.neurips.cc/paper/2021/hash/3b3fff6463464959dcd1b68d0320f781-Abstract.html) may provide a partial reason for the generator not memorizing training data. However, it is possible this may not fully reflect real-life GANs.
>
> We believe a complete theoretical justification for our counter-intuitive findings is very non-trivial given the under-developed state of GAN theory. Such a theoretical justification would have to explain the closeness of GAN samples to the test set while respecting the well-known limitations of GANs, such as mode collapse and samples having little benefits when used as data augmentation for training.
>
> However, we agree with the reviewer that finding a satisfying justification is extremely interesting and important. We hope our work inspires new theories for GANs that better fit their real-life properties.

---

### Official Review · Reviewer_1WB7 · 2021-11-03

**Correctness:** 4
**Technical Novelty And Significance:** 4
**Empirical Novelty And Significance:** 4
**Recommendation:** 8
**Confidence:** 3

**Main Review:**

Strengths: The paper is well-written, the results are important and convincing, and the related work is covered well.

Weaknesses: The results seem almost too good to be true. The GAN baseline not only outperforms other methods but blows them out of the water (see Table 1). Is there any explanation for how it was possible to improve so much over the previous competition winners? Some brief discussion of this might be helpful.

I want to make it clear that this is a minor weakness--I don't think the papers strong results should be held against it. I just wish there was a bit more discussion on this point.

**Summary Of The Paper:**

The paper considers the problem of predicting the test performance of neural networks, given only the training set and model hyperparameters. While there have been many methods proposed for this, the paper considers the simple baseline of generating synthetic test samples from a GAN and computing error using these samples. This baseline appears to outperform all or almost all existing methods on a variety of tasks, including a recent competition on predicting generalization.

**Summary Of The Review:**

This paper seems like a clear accept to me, unless there is some serious issue with the experiments that I missed. (I am not an expert in this field but they seemed quite convincing to me and basically speak for themselves.)

---

> ### Author Response · Authors · 2021-11-22
> **Response to Reviewer 1WB7**
>
> We thank the positive and encouraging comments by the reviewer!
>
> ### Q: Wish there was a bit more discussion about why this beats previous PGDL baselines by so much
>
> A: Honestly, we are also intrigued by the same question, and this is the motivation for our investigation presented in Section 6, which shows GAN samples appear closer to test samples in classifiers’ feature space compared to training samples.
>
> In fact, the paper [Arora et al 2017] proves the possibility of a generator that has mode collapse and yet can be used to predict the generalization of any bounded capacity classifier, provided the generator does not memorize training data. In fact, the recent empirical work on the difficulty of gradient inversion (https://proceedings.neurips.cc/paper/2021/hash/3b3fff6463464959dcd1b68d0320f781-Abstract.html) may provide a partial reason for the generator not memorizing training data. However,  this may not fully reflect real-life GANs.
>
> We have added the above discussion in the revision on page 2. We have also added that we think completely answering this question is non-trivial and requires a better understanding of GANs' strengths and limitations. We completely agree this is a fascinating future direction and hope our paper invites new efforts for analyzing/explaining GANs’ properties.

---

> > ### Comment · Reviewer_1WB7 · 2021-11-29
> > **Thanks!**
> >
> > Thank you, I appreciate your clarifications.

---

> > > ### Comment · Reviewer_d8Mz · 2021-12-01
> > > **Possible explanation**
> > >
> > > In the PGDL competition, the competitors do not know the dataset beforehand so they could not have used a pre-trained GAN to generate test data. I suppose data augmentation was meant as a proxy for the task. In real-world applications, however, you do know the dataset so the proposed method can easily be used.

---

### Author Response · Authors · 2021-11-22
**Revision uploaded**

The authors greatly appreciate the reviewers' valuable suggestions for improving our paper! We have uploaded a revision to reflect these suggestions. The modified parts are labeled in red.

---

### Decision · Program_Chairs · 2022-01-20

**Decision:**

Accept (Spotlight)

**Comment:**

The paper demonstrates that test error of image classification models can be accurately estimated using samples generated by a GAN. Surprisingly, this relatively simple proposed method outperforms existing approaches including ones from recent competitions. All reviewers agree this is a very interesting finding, even though theoretical analysis is lacking. Given the importance of the problem of predicting generalization, I recommend acceptance.